# From Representation to Action: A Unified Laplacian Framework for Spatial Representation and Path Planning

Junfeng Zuo [1 2 3 4 5]   Yuhang He [6]   Wen-Hao Zhang [7 8]   Fang Fang [1 2 3 5]   Si Wu [1 2 3 4 5]

## Abstract

Navigation in complex environments relies on internal spatial representations that guide action. While the brain employs a diverse repertoire of spatial tuning cells—including grid, place, and head-direction cells—a normative theory linking these static neural codes to the dynamic process of navigation remains elusive. In this work, we propose a Unified Laplacian Framework derived from first principles of representational smoothness and efficiency. We first demonstrate that diverse spatial codes emerge naturally as spectral decompositions of the Laplace operator. Crucially, bridging the gap from representation to action, we derive a computational-level navigation policy based on the Green's function potential. We show that this potential encodes the environment's intrinsic geometry to enable geometry-aware gradient ascent, achieving improved sample efficiency and generalization in goal-reaching tasks. Furthermore, we demonstrate that these spectral representations can be learned directly from high-dimensional visual inputs, supporting their learnability from sensory experience. Our results suggest that the "cognitive map" can be viewed as a spectral embedding of the Laplacian, providing a normative computational account that is biologically consistent with observed spatial-code phenomenology and useful for artificial agents.

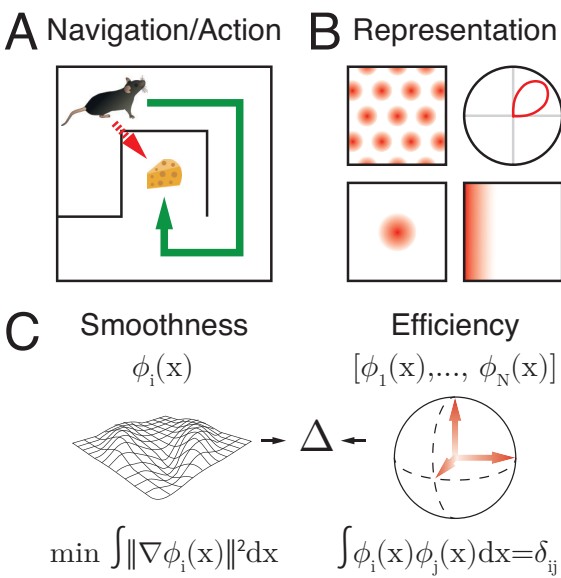

*Figure 1.* **A Unified Laplacian Framework for Spatial Cognition.** **A**. Navigation requires geometry-aware planning (green) rather than Euclidean greedy search (red). **B**. Canonical spatial tuning patterns observed in the brain. **C**. Representation principles: smoothness and efficiency.

[1]School of Psychology and Cognitive Sciences, Peking University. [2]IDG/McGovern Institute for Brain Research, Peking University. [3]Peking-Tsinghua Center for Life Sciences,Academy for Advanced Interdisciplinary Studies, Peking University. [4]Center of Quantitative Biology, Peking University. [5]Key Laboratory of Machine Perception (Ministry of Education), Peking University. [6]School of Mathematical Science, Peking University. [7]Lyda Hill Department of Bioinformatics, UT Southwestern Medical Center. [8]O'Donnell Brain Institute, UT Southwestern Medical Center.. Correspondence to: Si Wu <siwu@pku.edu.cn>.

*Proceedings of the 43rd International Conference on Machine Learning*, Seoul, South Korea. PMLR 306, 2026. Copyright 2026 by the author(s).

## 1. Introduction

The ability to navigate complex, non-convex environments is a hallmark of spatial intelligence (Fig. 1A). Over the past few decades, neuroscientists have systematically characterized the brain's "neural GPS", identifying a rich repertoire of spatial tuning cells in the hippocampal–entorhinal circuit (Fig. 1B), including place cells, grid cells, band cells, and boundary cells (O'Keefe, 1976; Hafting et al., 2005; Krupic et al., 2012; Solstad et al., 2008).

Existing computational theories have largely focused on the "mapping" aspect of this circuit—explaining how these diverse tuning patterns emerge from neural dynamics. Models based on continuous attractor networks (Burak & Fiete, 2009), dimensionality reduction (Dordek et al., 2016; Sorscher et al., 2023) or path integration (Xu et al., 2023; Chu et al., 2025) have successfully replicated the hexagonal

geometry of grid cells and the localized fields of place cells. However, these theories typically treat spatial codes as static coordinates for self-localization. The critical question of how these representations are mechanistically utilized to guide downstream action and path planning remains largely unaddressed.

This omission highlights a significant conceptual gap between "mapping" and "planning". In classical robotics and machine learning, these functions are traditionally compartmentalized: an agent first constructs a global map (e.g., via SLAM) and subsequently executes a separate, often computationally expensive search algorithm (e.g., Dijkstra or $A^*$) to compute a path. This separation introduces latency and fragility; if the environment changes or the agent is perturbed, the planner must often re-compute the solution from scratch. In contrast, biological navigation exhibits remarkable speed and flexibility. Animals can navigate to goals instantaneously and adapt to perturbations without apparent pauses for global re-planning. This suggests a more integrated architecture, where the spatial representation itself inherently encodes the navigational affordances required for control, eliminating the need for a separate search module.

In this work, we propose a **Unified Laplacian Framework** that bridges the gap. We posit that spatial codes can be understood not as arbitrary, but as solutions to a normative objective: minimizing Dirichlet energy (smoothness) under coding-efficiency constraints. We show that this objective naturally yields the eigenvalue problem of the Laplacian operator. By formalizing the cognitive map as a spectral eigenbasis (Belkin & Niyogi, 2003), we further demonstrate that the transition from representation to action is mathematically intrinsic: the static codes can be linearly combined to construct a Green's function potential that supports goal-directed navigation.

Our specific contributions are as follows:

- **Normative Derivation of Spatial Codes:** We show that minimizing the Dirichlet energy of a neural population yields the eigenfunctions of the Laplacian operator. In realistic bounded environments, this objective recovers tuning patterns consistent with key phenomenology of biological spatial cells.

- **The Green's Function Policy:** We derive a computational-level navigation mechanism based on the Laplacian Green's function. Unlike Euclidean vectors, which can induce wall-related local minima, the Green's function potential incorporates the environment's intrinsic geometry (Fig. 1A). Crucially, the agent instantaneously adjusts to perturbations by following the local gradient, without requiring global re-planning as in search-based methods (e.g., $A^*$).

- **Generalization and Learning Efficiency:** We demon-

strate that using the Green's function potential as an intrinsic reward improves the sample efficiency of reinforcement learning (RL) agents. In fixed-goal navigation, it accelerates convergence; in goal-conditioned tasks, it supports generalization across target locations where standard Euclidean shaping often fails.

By viewing the entorhinal cortex as a "Laplacian Eigen-Machine," our framework provides a mathematical bridge between the geometric structure of cognitive maps and navigation policies, offering a unified account of representation and action.

## 2. Laplacian Eigenfunctions as Spatial Representations

Here we focus on the spatial representation problem itself: the brain must encode continuous space into a finite neural code that is both stable and informative. We therefore ask what structural constraints a useful spatial representation should satisfy, and show that these constraints naturally lead to Laplacian eigenfunctions as the canonical basis.

### 2.1. Normative Constraints on Spatial Encoding

We model a spatial code as a population representation $\mathbf{r} : \Omega \to \mathbb{R}^N$, mapping each position $\mathbf{x} \in \Omega$ (e.g., $\mathbf{x} = (x, y)$ in a 2D environment) to a neural activity vector $\mathbf{r}(\mathbf{x}) = [r_1(\mathbf{x}), \ldots, r_N(\mathbf{x})]^\top$. We posit that to be computationally useful and consistent with biological constraints, this representation $\mathbf{r}(\mathbf{x})$ must satisfy two fundamental constraints: **smoothness** and **efficiency** (Fig. 1C).

**Smoothness via Dirichlet Energy Minimization.** A primary requirement for any spatial code is that the neural activity should vary smoothly across the physical space. This ensures that nearby locations have similar representations (locality preservation). This smoothness ensures that the representation is robust to sensory noise and enables generalization to unseen nearby locations (Malerba et al., 2025).

Intuitively, the smoothness of the $i$-th neuron's response corresponds to a low rate of change, or a small spatial gradient. We formalize this requirement using the total squared gradient magnitude across the domain, known as the *Dirichlet Energy* $\mathcal{E}(r_i)$:

$$\mathcal{E}(r_i) \triangleq \int_\Omega \|\nabla r_i(\mathbf{x})\|^2 d\mathbf{x}. \tag{1}$$

where $\nabla$ denotes the spatial gradient. We use this energy because it is local, quadratic, rotationally invariant, and generates diffusion on the domain. These properties make the later Green's-function construction natural: the same

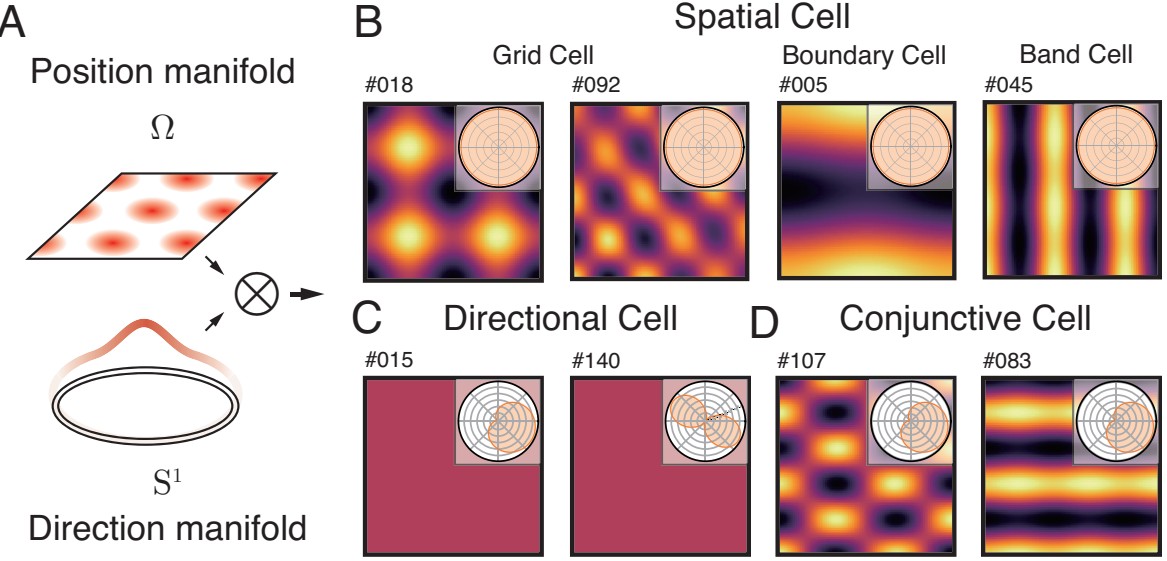

*Figure 2.* **Emergence of Spatial Tuning on the Pose Manifold. A.** State space is modeled as the product of the position manifold ($\Omega$) and the direction manifold ($S^1$). **B-D.** Eigenfunctions of the joint Laplacian naturally factorize into the full taxonomy of entorhinal cells: **(B)** pure spatial codes (Grid, Boundary, Band cells), **(C)** pure head-direction codes, and **(D)** conjunctive position-direction codes.

Laplacian that minimizes this smoothness cost also governs diffusion connectivity. This choice is not meant to exclude other smoothness principles, but it is the simplest one that preserves a linear spectral basis. Minimizing this functional penalizes rapid fluctuations, effectively acting as a low-pass filter to suppress high-frequency noises.

**Representation Efficiency via Normalization and Orthogonality.** Minimizing the energy alone yields a collapsed representation ($r_i(\mathbf{x}) = \text{const}$). To obtain non-trivial, efficient codes, we impose two constraints. First, we constrain each neuron's tuning function to have approximately unit norm across the space, capturing homeostatic regulation of its long-term activity budget. Second, to ensure that different neurons encode distinct features of the space (decorrelation) and cover the manifold efficiently, we require orthogonality between different basis functions. We combine these two constraints into a unified formulation:

$$\langle r_i, r_j \rangle_\Omega \triangleq \int_\Omega r_i(\mathbf{x}) r_j^\dagger(\mathbf{x}) \, d\mathbf{x} = \delta_{ij}, \qquad (2)$$

where $\delta_{ij}$ is the Kronecker delta ($\delta_{ij} = 1$ if $i = j$, else 0), and $r^\dagger$ denotes the complex conjugate ($r^\dagger = r$ for real-valued neural activities). Specifically, for a single unit $i$, the constraint simplifies to the unit norm condition $\int_\Omega ||r_i(\mathbf{x})||^2 d\mathbf{x} = 1$. Together, the smoothness and efficiency assumptions define the constrained objective $\min_{\{r_i\}} \sum_i \mathcal{E}(r_i)$ subject to Eq. (2).

## 2.2. Derivation of the Laplacian Representation

We now derive the optimal code under the smoothness objective in Eq. (1) and the efficiency constraints in Eq. (2). For each neuron $r_i$, we consider the Dirichlet energy with a unit-norm constraint via a Lagrange multiplier $\lambda_i$:

$$\mathcal{L}(r_i) = \int_\Omega \|\nabla r_i(\mathbf{x})\|^2 \, d\mathbf{x} - \lambda_i \left( \int_\Omega r_i(\mathbf{x})^2 \, d\mathbf{x} - 1 \right). \tag{3}$$

Applying calculus of variations (derivation in Appendix B), the stationary condition $\delta\mathcal{L} = 0$ yields the Laplacian eigenproblem:

$$-\Delta r_i(\mathbf{x}) = \lambda_i r_i(\mathbf{x}). \tag{4}$$

Therefore, the normative solution takes the form of Laplacian eigenfunctions. We denote the resulting optimal basis as $\{\phi_i\}_{i=1}^N$, with

$$r_i^*(\mathbf{x}) \equiv \phi_i(\mathbf{x}), \qquad -\Delta\phi_i = \lambda_i\phi_i, \tag{5}$$

where $\lambda_i$ indexes the spatial frequency (energy) of each mode.

## 3. Emergence of Spatial Coding Cells

In the last section, we established that the optimal spatial representation can be treated as the solution of the Laplacian eigenproblem ($-\Delta\phi = \lambda\phi$). We now investigate the specific forms these codes take in realistic environments.

### 3.1. Boundary Conditions

Solving the Laplacian eigenproblem requires specifying boundary behavior on $\partial\Omega$. In translationally invariant domains (e.g., $\mathbb{R}^2$) or periodic domains (e.g., $\mathbb{T}^2$), the eigenfunctions reduce to Fourier plane waves. Prior work has shown that additional biological constraints (e.g., non-negativity) can select hexagonal 2D patterns from these Fourier components (Sorscher et al., 2023; Dordek et al., 2016).

Real navigation, however, takes place in bounded enclosures. We therefore adopt the reflecting Neumann condition,

$$\frac{\partial\phi}{\partial\mathbf{n}}\Big|_{\partial\Omega} \equiv \nabla\phi(\mathbf{x})\cdot\mathbf{n}(\mathbf{x}) = 0, \qquad (6)$$

which prevents gradients from leaking across walls and makes the resulting tuning patterns sensitive to enclosure geometry. This naturally predicts distortions and rescaling of spatial codes in irregular environments (Krupic et al., 2015; Stensola et al., 2012).

### 3.2. Numerical Solutions on the Pose Manifold

To capture the agent's full state, we consider the pose manifold $\mathcal{M} = \Omega \times S^1$, where $\Omega$ is the physical enclosure and $S^1$ represents head direction. The spatial component inherits the Neumann condition in Eq. (6), while the angular dimension is naturally periodic. We approximate the continuous Laplacian on $\mathcal{M}$ by discretizing $\Omega$ with a grid graph and $S^1$ with a cycle graph (details in Appendix C).

This yields marginal Laplacians $L_{pos} \in \mathbb{R}^{d_p \times d_p}$ and $L_{head} \in \mathbb{R}^{d_h \times d_h}$, and a joint operator $L_{total} \in \mathbb{R}^{d \times d}$ on $\mathcal{M}$ given by the Kronecker sum

$$L_{total} = L_{pos} \oplus L_{head} \triangleq L_{pos} \otimes I_{d_h} + I_{d_p} \otimes L_{head}, \quad (7)$$

where $d = d_p d_h$ and $\otimes$ denotes the Kronecker product. Intuitively, the Kronecker sum is the Laplacian of the product pose space: $L_{pos}$ acts only on location, while $L_{head}$ acts only on heading. Because the two parts act on independent variables, their eigenfunctions multiply and their eigenvalues add. Sorting by the combined eigenvalue therefore naturally interleaves modes that are purely spatial, purely directional, or conjunctive, allowing one operator to generate the response-class taxonomy. We then compute the leading eigenfunctions of $L_{total}$, and reshape each eigenvector into a $d_p \times d_h$ tensor to obtain position-by-direction tuning curves. Formally, eigenpairs of the Kronecker sum satisfy $(\lambda_{i,j}, \phi_{i,j}) = (\lambda_i^{pos} + \lambda_j^{head}, \phi_i^{pos} \otimes \phi_j^{head})$.

### 3.3. Separation of Variables Induces Diverse Spatial Coding Cells

Building on the eigendecomposition of the joint operator $L_{total}$, the resulting modes admit a simple interpretation

on the product manifold $\mathcal{M} = \Omega \times S^1$: they separate into positional and directional components. Concretely, letting $\phi_i^{pos}(\mathbf{x})$ and $\phi_j^{head}(\theta)$ denote eigenfunctions on $\Omega$ and $S^1$, respectively, the joint modes can be written as

$$\phi_{i,j}(\mathbf{x},\theta) = \phi_i^{pos}(\mathbf{x})\,\phi_j^{head}(\theta), \qquad (8)$$

with associated eigenvalues $\lambda_{i,j}$ ordered by $\lambda_i^{pos} + \lambda_j^{head}$ (Eq. (7)). This separable structure directly organizes the numerically observed tuning patterns into familiar categories:

**Pure spatial cells** ($j = 0$). When $\phi_0^{head}(\theta)$ is constant, $\phi_{i,0}$ depends only on position and yields allocentric spatial patterns. Depending on domain geometry and mode frequency, these include band- and grid-like fields (Krupic et al., 2012); boundary-localized patterns can also be observed from low-frequency components under Neumann constraints (Fig. 2B).

**Pure head-direction codes** ($i = 0$). For $i = 0$, the joint mode reduces to a function of direction only: $\phi_{0,j}(\mathbf{x},\theta) \propto \phi_j^{head}(\theta)$ (Fig. 2C). Because $S^1$ is periodic, $\phi_j^{head}$ are Fourier modes (e.g., $\sin(k\theta)$ and $\cos(k\theta)$), yielding canonical unimodal head-direction tuning at low frequency and naturally generating multi-peaked variants at higher harmonics. Importantly, $k = 2$ harmonics naturally produce *antipodal* tuning with two peaks separated by $\pi$ (i.e., firing at two opposite headings, Fig. 2C right), matching experimental reports of *oppositely* bidirectional head-direction cells (Olson et al., 2017; Jacob et al., 2017).

**Conjunctive codes** ($i \neq 0$, $j \neq 0$). Mixed modes jointly modulate position and direction, yielding spatial fields gated by head direction (Fig. 2D), matching conjunctive grid×HD cells reported in entorhinal cortex (Sargolini et al., 2006).

## 4. From Representation to Navigation: The Green's Function Theory

We now address the missing link from *representation* to *action*: how can an agent use these static representations to generate goal-directed dynamics in environments with obstacles? This bridge is non-trivial because navigation is inherently global, and local similarity in representation space does not by itself determine a globally consistent direction toward a distant target in the physical space.

We take a normative viewpoint. A useful navigation mechanism should (i) respect the environment geometry, including obstacles, walls, and bottlenecks, without introducing spurious traps, and (ii) be computable from the representation itself. The Laplacian Green's function satisfies both: it is defined by the boundary-constrained Laplacian (hence topology-aware) and admits a closed-form expression by the Laplacian eigenfunctions $\{\phi_k\}$, yielding a simple policy

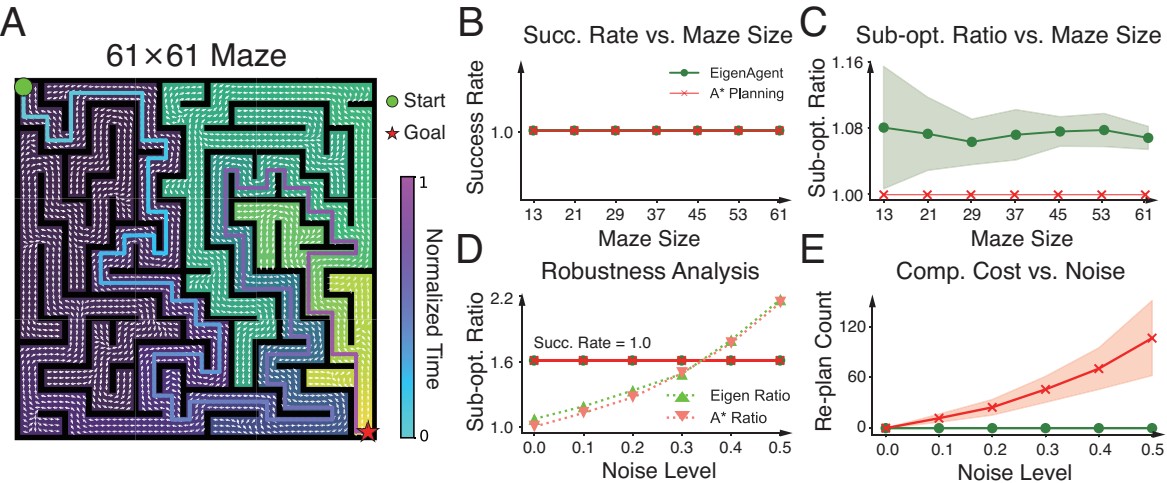

*Figure 3.* **Green's Function Policy Enables Robust Global Navigation. A**. The spectral potential induces a Green's-function gradient flow (white arrows) that successfully guides the agent around obstacles to the goal (star). **B-C**. The Green's function policy achieves 100% success rates with near-optimal path lengths across varying maze sizes, approaching the shortest-path reference computed by $A^*$ on the known maze graph. **D-E**. The Green's function policy is robust to noise with zero replanning cost, whereas $A^*$ computational overhead scales poorly with noise.

in which local gradient ascent induces a geodesic-like flow toward the goal.

For clarity and experimental simplicity, in this section we focus on positional navigation and restrict the state space to the spatial domain $\Omega$ (i.e., we omit the $S^1$ head-direction factor). Extending the Green's-function construction to the full pose manifold $\mathcal{M} = \Omega \times S^1$ follows directly from the product-Laplacian formulation in Sec. 3.2.

### 4.1. Potential-Based Navigation via the Green's Function

We formulate navigation on the manifold $\mathcal{M}$ as gradient flow on a potential field. Given a target location $\mathbf{x}_{tgt}$, let $U(\mathbf{x})$ denote a global potential defined over the state space, then the navigation policy can be defined as gradient ascent,

$$v(\mathbf{x}) \propto \nabla U(\mathbf{x}), \tag{9}$$

so that trajectories follow the steepest increase of potential $U$.

The key question is therefore: what potential $U$ provides a globally consistent gradient in the presence of obstacles? A naive Euclidean potential, e.g., $U_{\text{euc}}(\mathbf{x}) = -\|\mathbf{x} - \mathbf{x}_{tgt}\|$, ignores the manifold's intrinsic geometry and typically yields gradients that point into walls (Fig. 1A), producing invalid actions in non-convex environments (Kavraki et al., 2002).

To incorporate boundary constraints and encode the intrinsic geometry of the environment, we define $U$ as the solution to the Poisson equation on the manifold:

$$-\Delta U(\mathbf{x}) = \delta(\mathbf{x} - \mathbf{x}_{tgt}), \tag{10}$$

where $\Delta$ is the Laplace operator with boundary conditions imposed by obstacles, and $\delta$ is a Dirac delta source at the goal. By definition, the solution is the **Green's function** of the Laplacian, denoted $G(\mathbf{x}, \mathbf{x}_{tgt})$.

**Exact Green's function avoids interior traps.** Away from the goal, Eq. (10) reduces to $-\Delta U = 0$, i.e., $U$ is harmonic on $\Omega \setminus \{\mathbf{x}_{tgt}\}$. Harmonic functions admit no *interior* local maxima or minima, ruling out the obstacle-induced traps that plague Euclidean potentials. Consequently, the induced flow aligns with the intrinsic geometry encoded by the boundary-constrained Laplacian. This exact statement applies to the full continuous Green's function. In experiments, we use the finite spectral approximation $G_K$ with $K = 200$ modes; Appendix D.4 gives the sufficient residual condition and energy-capture analysis supporting its empirical robustness.

We thus define the *Green's Function Policy*:

$$v(\mathbf{x}) = \nabla_x G(\mathbf{x}, \mathbf{x}_{tgt}), \tag{11}$$

which is *reactive*—once $G(\cdot, \mathbf{x}_{tgt})$ is defined, action selection requires only local evaluation of the gradient at the current state, with no look-ahead horizon or explicit replanning.

### 4.2. Spectral Expansion and the Diffusion Perspective

Eq. (10) defines $G$ implicitly, but a neural circuit requires an explicit computation rule. Here we derive a closed-form expression of $G(\mathbf{x}, \mathbf{x}_{tgt})$ in the eigenbasis $\Phi$ from Sec. 2.2, which directly turns representation into action.

**Spectral derivation.** Since $\{\phi_k\}$ form a complete orthonormal basis, we expand $G(\mathbf{x}, \mathbf{x}_{tgt}) = \sum_k c_k\,\phi_k(\mathbf{x})$. Substituting into $-\Delta G = \delta$ and using $-\Delta\phi_k = \lambda_k\phi_k$ yields

$$\sum_k c_k\lambda_k\phi_k(\mathbf{x}) = \delta(\mathbf{x} - \mathbf{x}_{tgt}). \tag{12}$$

Projecting onto $\phi_k$ gives $c_k = \frac{1}{\lambda_k}\phi_k(\mathbf{x}_{tgt})$, and therefore the spectral expansion[1]:

$$G(\mathbf{x}, \mathbf{x}_{tgt}) = \sum_{k=1}^{\infty} \frac{1}{\lambda_k}\phi_k(\mathbf{x})\phi_k(\mathbf{x}_{tgt}). \tag{13}$$

Eq. (13) provides an explicit computation rule: given eigenfeatures $\{\phi_k(\mathbf{x})\}$ and the goal code $\{\phi_k(\mathbf{x}_{tgt})\}$, the potential is obtained by a mode-wise product followed by a fixed linear readout with weights $1/\lambda_k$. This closes the representation–action loop using the same basis.

**Gradient computation.** The policy depends on $\nabla_{\mathbf{x}}G(\mathbf{x}, \mathbf{x}_{tgt})$, which can also be obtained by the spectral expansion:

$$\nabla_{\mathbf{x}}G(\mathbf{x}, \mathbf{x}_{tgt}) = \sum_{k=1}^{\infty} \frac{1}{\lambda_k}\,\phi_k(\mathbf{x}_{tgt})\,\nabla_{\mathbf{x}}\phi_k(\mathbf{x}). \tag{14}$$

Thus, the gradient field is a *local* linear combination of $\{\nabla\phi_k(\mathbf{x})\}$ evaluated at the agent's current state, with goal-dependent coefficients $\phi_k(\mathbf{x}_{tgt})/\lambda_k$. In particular, once the goal is fixed, these coefficients are constant across space, and action selection reduces to a local weighted summation.

**Spectral Tilt and Multi-scale Planning.** The term $1/\lambda_k$ acts as a low-pass filter ('Spectral Tilt'), amplifying the contribution of low-frequency (large-scale) modes. This theoretically explains the functional hierarchy of the grid cell system: large-scale grids (small $\lambda_k$) dominate the global potential $G$, enabling long-range path planning, while fine-scale grids (large $\lambda_k$) provide local detail but have minimal impact on the global gradient.

**Integration of Time Horizon.** This spectral form also connects to the dynamic perspective of exploration. The Green's function is mathematically equivalent to the temporal integration of the Diffusion Kernel $K_t(\mathbf{x}, \mathbf{y})$ (describing Brownian motion) (Coifman & Lafon, 2006):

$$G(\mathbf{x}, \mathbf{x}_{tgt}) = \int_0^{\infty} \left(\sum_{k=1}^{\infty} e^{-\lambda_k t}\phi_k(\mathbf{x})\phi_k(\mathbf{x}_{tgt})\right) dt \tag{15}$$

---

[1]With Neumann boundaries, the Green's function is defined up to an additive constant (and the constant mode is omitted in the spectral expansion). Since the policy depends only on $\nabla G$, this gauge freedom does not affect behavior.

Physically, this means that ascending $G$ follows directions that maximize *diffusion connectivity* aggregated across all time horizons. Unlike methods that require selecting a specific look-ahead horizon $T$, the Green's function yields a horizon-free notion of reachability by integrating contributions from both fast (local) and slow (global) diffusion processes, thereby providing robust paths through the environment's intrinsic geometry (details in Appendix D).

### 4.3. Global Navigation via Green's Function Gradient

We test the central prediction of Sec. 4.1: whether the Green's-function potential induces a reactive policy that respects obstacles, enabling global navigation. We evaluate an `EigenAgent` equipped with the Green's function policy in non-convex maze environments (Fig. 3A) and compare against $A^*$ as a shortest-path reference on the same known discrete maze graph, with complete map knowledge and an admissible Euclidean heuristic. This comparison benchmarks path quality, whereas the Green's-function policy emphasizes reactive execution once the potential is available.

#### 4.3.1. GLOBAL CONSISTENCY: NO SPURIOUS TRAPS AND NEAR-OPTIMAL PATHS

Fig. 3A visualizes a $61 \times 61$ maze and the induced Green's function potential in it. The potential landscape forms a smooth global gradient flow toward the goal, and the resulting streamlines bend around obstacles rather than pointing into walls, consistent with a geometry-aware potential defined by the boundary-constrained Laplacian.

Across maze sizes (Fig. 3B-C), `EigenAgent` attains 100% success rate, matching the known-graph $A^*$ reference. Moreover, the resulting trajectories are close to optimal: the sub-optimality ratio (Path Length / Optimal Length) stabilizes around $1.08$, especially when the maze is large, indicating that the Green's-function gradient provides an accurate, geometry-aware solution for shortest-path navigation.

#### 4.3.2. ROBUSTNESS WITHOUT REPLANNING

We next perturb the agent with random motion noise $\epsilon \in [0, 0.5]$ to test robustness under stochastic deviations. Specifically, at each time step the agent is displaced to a nearby state with probability $\epsilon$, mimicking incidental perturbations in realistic settings. While noise increases path length for both methods (Fig. 3D), `EigenAgent` degrades gracefully and remains comparable to the $A^*$ baseline.

Crucially, the computational burden of `EigenAgent` is significantly lower. Because $A^*$ algorithm must recompute plans after perturbations, its replanning count grows rapidly with noise (Fig. 3E). In comparison, `EigenAgent` incurs

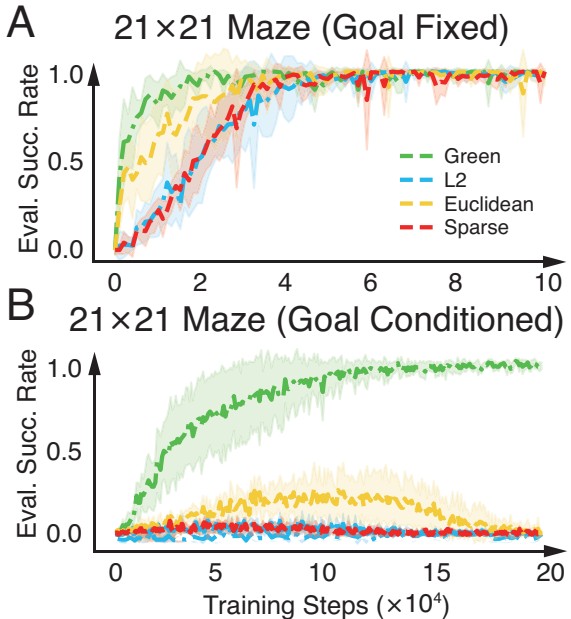

**A** 21×21 Maze (Goal Fixed)

**B** 21×21 Maze (Goal Conditioned)

*Figure 4.* **Green's-function reward shaping accelerates learning and supports goal-conditioned generalization. A**. In the fixed-goal $21 \times 21$ maze, Green's-function shaping reaches near-perfect success substantially faster than the Euclidean, $L_2$ spectral-distance, and sparse-reward baselines. **B**. In the goal-conditioned $21 \times 21$ random-maze setting, training is shown over $5 \times 10^5$ steps. Green's-function shaping rapidly approaches near-perfect test success and remains high, whereas Euclidean shaping shows only transient partial improvement and the $L_2$ spectral-distance and sparse-reward baselines remain near zero.

zero replanning: it simply evaluates the local gradient at the perturbed state, maintaining a constant per-step complexity of $\mathcal{O}(k)$ over the eigenmodes. This illustrates a key advantage of Green's-function navigation: global consistency is inherent to the representation, while action selection remains purely local and reactive.

## 4.4. Learning to Navigate with Laplacian Representations

The analysis in Sec. 4 suggests a concrete navigational role for Laplacian representations beyond encoding. Here, we validate this hypothesis in a downstream RL setting. Specifically, we investigate whether (i) Laplacian eigenfunctions can serve as effective spatial representations, and (ii) the Green's-function potential can act as a geometry-aware shaping signal that accelerates learning under sparse rewards and supports generalization across changing goals.

### 4.4.1. EXPERIMENTAL SETUP

We train an MLP-based agent in a $21 \times 21$ maze using Q-learning. The agent receives a Laplacian representation of its current state and goal, $s_t = [\{\phi_k(\mathbf{x}_t)\}, \{\phi_k(\mathbf{x}_{tgt})\}]$, where

$\{\phi_k\}$ contains the first $k = 200$ Laplacian eigenfunctions.

To mitigate sparse rewards, we use *potential-based reward shaping* (Ng et al., 1999): given a shaping potential $\Psi(\cdot)$, we add the shaping term

$$F(s, s') = \gamma \Psi(s') - \Psi(s) \qquad (16)$$

to the environment reward at every step. We compare four shaping potentials:

1. **Green's function:** $\Psi(\mathbf{x}) = G(\mathbf{x}, \mathbf{x}_{tgt})$;

2. **Euclidean potential:** $\Psi(\mathbf{x}) = -\|\mathbf{x} - \mathbf{x}_{tgt}\|_2$;

3. $L_2$ **spectral distance:** $\Psi(\mathbf{x}) = -\|\phi(\mathbf{x}) - \phi(\mathbf{x}_{tgt})\|_2$;

4. **Sparse reward:** no shaping.

The $L_2$ spectral-distance baseline uses the same Laplacian eigenfunctions as the Green potential. Because methods based on successor representations share this eigenbasis, it serves as a same-representation comparison rather than a separate planner. We evaluate performance using the success rate of reaching the goal, reported during training in Fig. 4. These navigation and RL experiments assume that the spectral basis is available from the known maze graph; Sec. 5 tests whether related low-frequency geometry can be recovered from raw visual experience, but does not yet close the full pixel-to-action loop. Note that potential-based shaping uses a potential *difference* across successive states up to discount. When state transitions are small, this difference acts as a local approximation to the *potential gradient*, aligning the RL learning signal with the gradient-ascent control view in Sec. 4.1.

### 4.4.2. RESULTS: ACCELERATED LEARNING AND GOAL-CONDITIONED GENERALIZATION

**Goal-fixed navigation.** With a fixed goal (Fig. 4A), Green's shaping converges substantially faster, reaching near 100% success within $2 \times 10^4$ steps. Euclidean shaping eventually learns but exhibits a slower start, consistent with the fact that Euclidean distance provides an unreliable progress signal in the presence of obstacle-induced topology.

**Goal-conditioned generalization.** The benefit of geometry-aware shaping becomes most pronounced when both start and goal are randomized in each episode (Fig. 4B). Over $5 \times 10^5$ training steps, Green's-function shaping rapidly approaches near-perfect test success and remains high throughout training. Euclidean shaping shows only transient partial improvement, while the $L_2$ spectral-distance and sparse-reward baselines remain near zero. This result sharpens the comparison: the Green potential provides a boundary-constrained, geometry-aware progress signal that transfers

across changing start–goal pairs, whereas Euclidean distance and uniformly weighted spectral distance do not provide reliable goal-conditioned shaping.

Overall, these results show that Laplacian representations may support navigation not only as a coordinate-like representation, but also as a mechanism for constructing an intrinsic geometry-aware potential that guides exploration and planning in complex environments.

## 5. Learning Spatial Representations from Visual Inputs

The analytical derivations above are grounded in explicit coordinates. However, a critical test of our framework is whether biological circuits can recover these spectral representations directly from **high-dimensional sensory streams** (e.g., visual inputs) without privileged access to ground-truth coordinates or a predefined state-space graph.

To demonstrate this, we trained an agent equipped with a CNN encoder in a photorealistic 3D environment (`MemoryMaze` Open Field) (Pasukonis et al., 2022). The agent explored via random diffusion (Fig. 1A), receiving *raw egocentric RGB observations* ($64 \times 64$) with no positional supervision (Fig. 5B). The network was trained to approximately enforce representational smoothness by minimizing the variance of neural activity across time steps (Wiskott & Sejnowski, 2002), together with variance and covariance regularization to promote coding efficiency.

Despite the absence of ground-truth coordinates, the network spontaneously recovered spatial features characteristic of Laplacian eigenfunctions (Fig. 5C), including band-like and grid-like tuning patterns. Moreover, a linear probe on frozen embeddings decoded the agent's position with high accuracy, reinforcing the view that Laplacian-like spatial codes can emerge robustly from perceptual experience under unsupervised learning.

## 6. Discussion

We have proposed a unified framework where spatial representation and navigation are dual aspects of the same spectral operator. The Laplacian eigenfunctions serve as the optimal localization system (the representation), while the Green's function acts as the optimal flow field (the action). Here, we address the computational-level interpretation and biological scope of these computations, as well as their unification of geometric and predictive theories.

### 6.1. Computational-Level Interpretation and Biological Scope

While our analytical model relies on numerical eigensolvers, one possible biological interpretation is that neural circuits

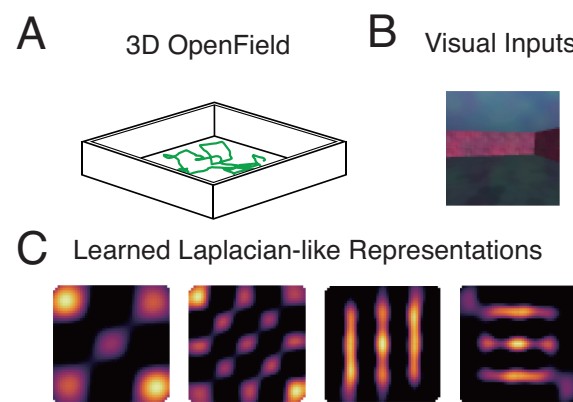

*Figure 5.* **Learning Laplacian Representation from Sensory Inputs. A-B**. Agent explores a 3D environment receiving only raw egocentric visual inputs. **C**. The network spontaneously recovers Laplacian-like spatial representations directly from sensory experience.

approximate these solutions through local synaptic plasticity in two stages.

First, sensory inputs are likely mapped into a high-dimensional feature space, a process paralleling *Random Fourier Features* (RFF) (Rahimi & Recht, 2007). This effectively linearizes the environment's transition manifold in a high-dimensional space, producing a neural approximation of the diffusion kernel directly from sensory streams. Second, acting upon these projected features, downstream circuits can extract spectral embeddings using simple Hebbian mechanisms. Algorithms such as *Sanger's Rule* enable networks to perform online PCA (Oja, 1982; Sanger, 1989); when applied to the RFF-projected signals generated during random exploration, this dynamics could recover the Laplacian eigenfunctions. Thus, the grid cell system may act as a spectral analyzer of the environmental diffusion operator. Future work must investigate how these representations adapt to changing boundaries via rapid synaptic re-weighting or attractor dynamics.

### 6.2. Unifying Geometric and Predictive Maps

Our framework unifies the geometric view of the hippocampus with established theories in predictive coding and reinforcement learning (RL). The "Predictive Map" hypothesis posits that place cells encode the Successor Representation (SR) to predict future occupancy (Stachenfeld et al., 2017). Similarly, in reinforcement learning, the "Proto-value Function" framework utilizes Laplacian eigenfunctions as an efficient basis for value function approximation (Mahadevan & Maggioni, 2007).

We demonstrate that these approaches are not distinct strategies but emerge from the same normative principle: the minimization of Dirichlet energy under efficiency constraints.

Furthermore, we explicitly link these discrete RL constructs to the continuous physics of diffusion. The SR matrix $M = (I - \gamma T)^{-1}$ implicitly encodes graph connectivity. We identify our Green's function derivation as the continuous, "horizon-free" limit of this map. As $\gamma \to 1$ and time-steps shrink, the operator $(I - T)$ converges to the normalized Laplacian $L$, and the SR matrix converges to the Green's function:

$$\lim_{\gamma \to 1, \Delta t \to 0} M \propto L^{-1} \equiv G \qquad (17)$$

This derivation reveals that the "Cognitive Map", "Successor Representation", and "Proto-value Functions" are dual spectral descriptions of the same object. Our specific contribution lies in demonstrating that this predictive object $G$ is not merely a statistical dependency, but a physically interpretable potential field. Crucially, this perspective allows us to derive a geometry-aware navigation policy directly from the representation's gradient, thereby bridging the gap between static representation and active planning.

## 7. Conclusion

In this work, we presented a unified framework that derives the diverse phenomenology of entorhinal spatial cells from first principles of representational smoothness and coding efficiency. We demonstrated that these normative constraints naturally yield the eigenfunctions of the Laplacian operator, recovering the taxonomy of grid, band, and boundary cells observed *in vivo* without relying on *ad hoc* geometric assumptions.

Beyond representation, we elucidated the computational utility of these codes by deriving a computational-level navigation policy based on the Laplacian Green's function. This construction bridges the gap between static spatial maps and dynamic action, enabling geometry-aware path planning in complex environments.

Furthermore, we evaluated the framework in high-dimensional sensory domains. We showed that these spectral representations can emerge directly from raw visual streams, suggesting a biologically consistent route for extracting global geometric structure from local sensory experience.

Ultimately, our results suggest that the entorhinal-hippocampal circuit can be interpreted computationally as a "Laplacian Eigen-Machine". By encoding space into Laplacian eigen basis, this view transforms the complex, non-convex problem of path planning into a simple linear operation, unifying mapping, localization, and action into a single, elegant computational framework.

## Impact Statement

This paper presents work whose goal is to advance the field of Machine Learning. There are many potential societal consequences of our work, none which we feel must be specifically highlighted here.

## Acknowledgement

This work was supported by the National Natural Science Foundation of China (no. T2421004 to S.W.), the National Key Research and Development Program of China (2024YFF1206500), the Science and Technology Innovation 2030-Brain Science and Brain-inspired Intelligence Project (no. 2021ZD0200204, S.W.).

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

## A. Related Work

**Normative Spatial Representations.** While early models focused on mechanistic implementations, e.g., continuous attractor neural networks (CANN) (Burak & Fiete, 2009), recent normative theories derive grid codes via dimensionality reduction (Dordek et al., 2016) or path integration optimization (Banino et al., 2018; Sorscher et al., 2023; Chu et al., 2025; Xu et al., 2023; Cueva & Wei, 2018). However, these works typically treat spatial codes as static coordinates for localization. Our framework extends this normative approach—specifically Dirichlet energy minimization—to address the *utility* of these codes, deriving a navigation policy directly from the representation's spectral structure.

**Spectral Methods and Predictive Maps.** Our work unifies the "Predictive Map" hypothesis (Stachenfeld et al., 2017) and Proto-value Functions (Mahadevan & Maggioni, 2007) by formally identifying the Green's function as the continuous, horizon-free limit of the Successor Representation (Eq. (17)). Unlike prior spectral RL (Wu et al., 2018; Machado et al., 2018) methods that use eigenfunctions merely as basis features for learning policies, we utilize the Green's function potential to define an intrinsic, geometry-aware gradient flow, eliminating the need for extensive policy training.

**Geometry-Aware Planning.** Classical navigation often separates mapping from planning (e.g., A*) or suffers from local minima in potential fields (Kavraki et al., 2002). While harmonic functions avoid such traps (Connolly & Grupen, 1993), solving partial differential equations online is biologically implausible. We propose a spectral solution: by expanding the potential in the Laplacian eigenbasis, we reduce global path planning to a simple linear summation at the computational level that respects environmental geometry without explicit search.

## B. Derivation of the Laplacian Representation

In Sec. 2, we posited that the optimal spatial codes emerge from minimizing the Dirichlet energy functional (Eq. 1) under the unit-norm constraint (Eq. 2). Here, we provide the formal derivation using the calculus of variations.

We define the Lagrangian functional $\mathcal{L}(\phi)$ for a single neural representation $\phi(\mathbf{x})$ with a Lagrange multiplier $\lambda$:

$$\mathcal{L}(\phi) = \int_\Omega ||\nabla\phi(\mathbf{x})||^2 \, d\mathbf{x} - \lambda \left( \int_\Omega \phi(\mathbf{x})^2 \, d\mathbf{x} - 1 \right) \tag{18}$$

To find the stationary functions, we apply the Euler-Lagrange equation for the functional density $F(\mathbf{x}, \phi, \nabla\phi) = ||\nabla\phi||^2 - \lambda(\phi^2 - 1)$:

$$\frac{\partial F}{\partial \phi} - \nabla \cdot \frac{\partial F}{\partial(\nabla\phi)} = 0 \tag{19}$$

Computing the partial derivatives: 1. The derivative with respect to the scalar field is $\frac{\partial F}{\partial \phi} = -2\lambda\phi(\mathbf{x})$. 2. The derivative with respect to the gradient vector is $\frac{\partial F}{\partial(\nabla\phi)} = 2\nabla\phi(\mathbf{x})$.

Substituting these terms back into Eq. (19) yields:

$$-2\lambda\phi(\mathbf{x}) - \nabla \cdot (2\nabla\phi(\mathbf{x})) = 0 \tag{20}$$

Dividing by $-2$ and utilizing the definition of the Laplace operator $\Delta = \nabla \cdot \nabla$, we recover the Helmholtz equation:

$$-\Delta\phi(\mathbf{x}) = \lambda\phi(\mathbf{x}) \tag{21}$$

This confirms that the stationary points of the smoothness objective are the eigenfunctions of the negative Laplacian, where the Lagrange multiplier $\lambda$ corresponds to the eigenvalue (spatial frequency).

## C. Numerical Implementation Details

In this section, we detail the discretization of the continuous Laplacian operator on the pose manifold $\mathcal{M} = \Omega \times S^1$ and the numerical enforcement of boundary conditions.

**C.1. Boundary Conditions and Graph Construction**

To numerically approximate the Laplace operator, we discretize the domain as a graph and compute the graph Laplacian. The specific choice of graph connectivity implicitly enforces the boundary conditions.

**Spatial Domain (Neumann Condition).** For the spatial domain $\Omega$, we enforce the Neumann boundary condition ($\nabla\phi \cdot \mathbf{n} = 0$). We construct a grid graph $\mathcal{G}_{pos} = (V_{pos}, E_{pos})$ with $d_p$ nodes. The discrete Laplacian is defined as $L_{pos} = D_{pos} - A_{pos}$, where $A_{pos}$ is the adjacency matrix and $D_{pos}$ is the degree matrix. Crucially, nodes at the boundaries of the grid have fewer neighbors (lower degree) and no edges connecting to an external sink. This construction mathematically corresponds to a zero-flux condition across the boundary, providing a discrete approximation of Eq. (6) without explicit constraint handling.

**Directional Domain (Periodic Condition).** For the head-direction manifold $S^1$, we enforce periodic boundary conditions. We construct a cycle graph $\mathcal{G}_{dir}$ with $d_h$ nodes. The periodicity is enforced by adding an edge between the first node (0) and the last node ($d_h - 1$) in the adjacency matrix.

**C.2. Joint Laplacian via Kronecker Sum**

To model the agent on the full pose manifold, we construct the total Laplacian $L_{total} \in \mathbb{R}^{d\times d}$ (where $d = d_p \times d_h$). Assuming the transition dynamics in space and direction are independent, the joint operator is given by the Kronecker sum of the marginal Laplacians:

$$L_{total} = L_{pos} \oplus L_{head} \triangleq L_{pos} \otimes I_{d_h} + I_{d_p} \otimes L_{head} \tag{22}$$

where $\otimes$ denotes the Kronecker product and $I$ is the identity matrix.

**C.3. Eigendecomposition and Tensor Structure**

We solve the eigenvalue problem $L_{total}\phi_k = \lambda_k\phi_k$. A key property of the Kronecker sum is that the eigenvalues and eigenvectors of $L_{total}$ factorize into those of the marginal operators. Specifically, for any pair of spatial eigenmode $(\lambda_i^{pos}, \phi_i^{pos})$ and directional eigenmode $(\lambda_j^{head}, \phi_j^{head})$, there exists a joint eigenpair:

$$\lambda_k = \lambda_i^{pos} + \lambda_j^{head}, \qquad \phi_k = \phi_i^{pos} \otimes \phi_j^{head} \tag{23}$$

Implementation-wise, we verify this by computing the eigendecomposition of the sparse matrix $L_{total}$ directly. The resulting eigenvectors $\phi_k \in \mathbb{R}^d$ are then reshaped into tensors of size $d_p \times d_h$ to recover the position-by-direction tuning curves visualized in the main text.

# D. Derivation of the Green's Function Policy

In Sec. 4, we introduced the Green's function policy for navigation. Here, we provide the mathematical derivation linking the static Laplacian representations to dynamic diffusion processes and the resulting potential field.

**D.1. From Diffusion to the Heat Kernel**

We model the agent's stochastic exploration as a diffusion process governed by the heat equation $\frac{\partial p}{\partial t} = \Delta p$. By expanding the probability density $p(\mathbf{x}, t)$ in the basis of Laplacian eigenfunctions $\{\phi_k\}$, the solution for an agent starting at $\mathbf{y}$ is given by the Heat Kernel $K_t(\mathbf{x}, \mathbf{y})$:

$$K_t(\mathbf{x}, \mathbf{y}) = \sum_{k=1}^{\infty} e^{-\lambda_k t}\phi_k(\mathbf{x})\phi_k(\mathbf{y}) \tag{24}$$

where $\lambda_k$ and $\phi_k$ are the eigenvalues and eigenfunctions derived in Appendix B. This kernel represents the transition probability density between states $\mathbf{y}$ and $\mathbf{x}$ over a time horizon $t$.

## D.2. Derivation of the Green's Function

To obtain a time-invariant navigation potential, we integrate the diffusion connectivity over all timescales. The Green's function $G(\mathbf{x}, \mathbf{y})$ is defined as the temporal integration of the Heat Kernel (centered to handle the zero mode):

$$G(\mathbf{x}, \mathbf{y}) \triangleq \int_0^\infty \left( K_t(\mathbf{x}, \mathbf{y}) - \frac{1}{|\Omega|} \right) dt \tag{25}$$

Substituting the spectral expansion from Eq. (24) and using the identity $\int_0^\infty e^{-\lambda_k t} dt = \frac{1}{\lambda_k}$ (for $\lambda_k > 0$), we recover the spectral definition of the Green's function used in the main text:

$$G(\mathbf{x}, \mathbf{y}) = \sum_{k=1}^\infty \frac{1}{\lambda_k} \phi_k(\mathbf{x}) \phi_k(\mathbf{y}) \tag{26}$$

This confirms that the Green's function is the pseudo-inverse of the Laplacian operator on the spectral domain.

## D.3. Gradient Computation for Navigation

The navigation policy is defined as gradient ascent on this potential: $v(\mathbf{x}) \propto \nabla_x G(\mathbf{x}, \mathbf{x}_{tgt})$. Since the gradient operator acts linearly on the spatial coordinates, we can compute it term-wise from the spectral expansion:

$$\nabla_x G(\mathbf{x}, \mathbf{x}_{tgt}) = \sum_{k=1}^\infty \frac{\phi_k(\mathbf{x}_{tgt})}{\lambda_k} \nabla \phi_k(\mathbf{x}) \tag{27}$$

This formula provides the explicit computation rule for the "EigenAgent": the global navigation vector is a weighted sum of the local gradients of each eigenfunction ($\nabla \phi_k(\mathbf{x})$), where the weights are determined by the target's representation and the inverse eigenvalues.

## D.4. Finite Spectral Truncation

The exact Green's function in Eq. (26) uses the full Laplacian spectrum. The implementation instead uses the first $K$ non-constant modes:

$$G_K(\mathbf{x}, \mathbf{y}) = \sum_{k=1}^K \frac{1}{\lambda_k} \phi_k(\mathbf{x}) \phi_k(\mathbf{y}). \tag{28}$$

Away from the goal $\mathbf{y}$, applying the Laplacian to $G_K$ gives

$$\Delta G_K(\mathbf{x}, \mathbf{y}) = \frac{1}{|\Omega|} + \rho_K(\mathbf{x}), \quad \rho_K(\mathbf{x}) = \sum_{k=K+1}^\infty \phi_k(\mathbf{x}) \phi_k(\mathbf{y}). \tag{29}$$

Thus the exact Green's-function limit has a positive subharmonicity margin $1/|\Omega|$, while truncation perturbs this margin by the omitted spectral tail $\rho_K$. A sufficient condition for preserving strict subharmonicity away from the goal is

$$\sup_{\mathbf{x} \neq \mathbf{y}} |\rho_K(\mathbf{x})| < \frac{1}{|\Omega|}. \tag{30}$$

Under this condition, $G_K$ avoids interior local maxima away from the goal. This is a sufficient condition rather than a complete characterization of all finite spectral approximations.

The $1/\lambda_k$ weighting also makes the potential converge rapidly in energy. Using the spectral energy of the full Green's function as reference, the retained modes capture the potential energy shown in Table 1.

As shown in Table 1, $K = 200$ already captures 99.7% of the potential energy. This explains why the finite spectral approximation preserves the large-scale geometry needed for navigation, while the exact no-interior-trap statement remains a property of the full Green's-function limit.

*Table 1.* Potential energy captured by the truncated Green's function. These values support empirical finite-$K$ robustness but do not replace the exact infinite-basis result.

| $K$ | POTENTIAL ENERGY CAPTURED |
|---|---|
| 50 | 98.8% |
| 100 | 99.4% |
| 200 | 99.7% |
| 500 | 99.9% |

# E. Visual Learning Implementation Details

In Sec. 5, we demonstrated that grid and band cells emerge from self-supervised learning on visual inputs. Here we detail the architecture and training protocol based on the `MemoryMaze` environment.

### E.1. Environment and Data Collection

We used the `MemoryMaze-OpenField-9x9-ExtraObs-v0` environment, a high-fidelity continuous control task based on the MuJoCo physics engine.

- **Visual Input:** The input state $\mathbf{s}_t \in \mathbb{R}^{9 \times 64 \times 64}$ consists of a stack of the $k = 3$ most recent RGB frames, resized to $64 \times 64$. This frame stacking allows the network to implicitly infer velocity and heading from static images.

- **Isotropic Diffusion:** To satisfy the theoretical requirement that the temporal transition operator approximates the Laplacian, the agent must explore the manifold uniformly. We utilized an `OracleDiffusionActor` which follows a correlated random walk (Ornstein-Uhlenbeck process on heading, $\sigma = 0.6$) with randomized boundary repulsion, ensuring dense coverage of the arena.

### E.2. Network Architecture

The architecture (Fig. 6) consists of a Convolutional Encoder and an MLP Projector.

**CNN Encoder:** A 4-block network maps the high-dimensional visual input to a compact latent embedding $\mathbf{h}_t \in \mathbb{R}^{128}$. Each block contains a Convolution, Batch Normalization, and GELU activation.

The final feature map ($256 \times 4 \times 4$) is flattened to dimension 4096 and projected linearly to the encoder output dimension of 128.

**Projector:** A 3-layer MLP ($128 \rightarrow 1024 \rightarrow 1024 \rightarrow 64$) maps the representation to the final embedding $\mathbf{z}_t$ where the loss is applied. This projector is critical for decorrelating the feature variables during training.

### E.3. Training Objective: Temporal VICReg

We trained the network to minimize the **Temporal VICReg** loss (Bardes et al., 2021):

$$\mathcal{L} = \lambda \sum_t \|\mathbf{z}_t - \mathbf{z}_{t+k}\|^2 + \mu \sum_j \max(0, 1 - \text{std}(z^j)) + \nu \sum_{j \neq l} \text{Cov}(\mathbf{z})^2_{jl} \tag{31}$$

**Geometric $k$-step Sampling.** A key innovation in our experiment (implemented in `replay.py`) is the sampling of the time gap $k$. Instead of a fixed horizon, we sample $k$ from a geometric distribution ($p = 0.5$, clipped to $k_{max} = 5$). This

*Table 2.* **CNN Encoder Specification.**

| Layer | In Channels | Out Channels | Kernel | Stride |
|---|---|---|---|---|
| Conv 1 | 9 | 32 | $5 \times 5$ | 2 |
| Conv 2 | 32 | 64 | $3 \times 3$ | 2 |
| Conv 3 | 64 | 128 | $3 \times 3$ | 2 |
| Conv 4 | 128 | 256 | $3 \times 3$ | 2 |

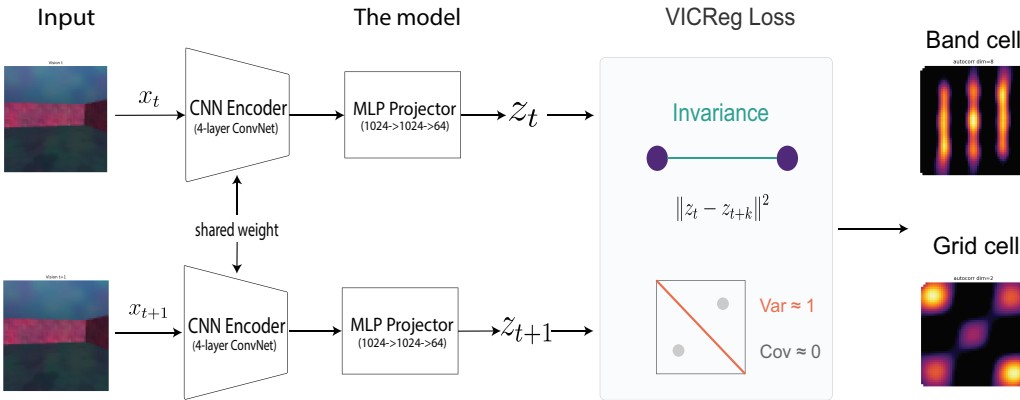

*Figure 6.* **Network Architecture.** The model processes stacked visual frames to learn a low-dimensional spectral embedding via the Temporal VICReg objective.

ensures the network learns features that are invariant across multiple timescales, effectively integrating the diffusion operator to capture global geometry.

**Data Augmentation.** To prevent the network from relying on low-level pixel statistics (e.g., lighting artifacts), we applied RandomShift ($\pm 2$ pixels) and IntensityJitter ($0.95 - 1.05$) independently to the source and target frames.

**Hyperparameters.** We used the AdamW optimizer with learning rate $5 \times 10^{-4}$ and weight decay $10^{-6}$. The batch size was 256, and training ran for $50,000$ steps. The loss coefficients were set to $\lambda = 25.0$ (Smoothness), $\mu = 25.0$ (Variance), and $\nu = 1.0$ (Covariance).

