# OpenReview forum: "From Representation to Action: A Unified Laplacian Framework for Spatial Representation and Path Planning"
_ICML.cc/2026/Conference — ICML 2026 regular_

### Official Review · Reviewer_Hg76 · 2026-03-10

**Soundness:** 3
**Presentation:** 3
**Significance:** 3
**Originality:** 3
**Overall Recommendation:** 4
**Confidence:** 4

**Summary:**

The paper proposes a Unified Laplacian Framework that links spatial representations observed in the hippocampal–entorhinal circuit to goal-directed navigation. Starting from two normative principles (smoothness via Dirichlet energy minimization, and coding efficiency via orthonormality), the authors show that the optimal neural population code solves the Laplacian eigenvalue problem. They then demonstrate that diverse spatial cell types (grid, boundary, band, and head-direction cells) emerge naturally as eigenfunctions of the joint position-direction Laplacian. The same eigenbasis is used to construct a Green's function navigation potential, whose gradient yields a trap-free, reactive policy. The framework is validated in maze navigation experiments against A*, in RL reward-shaping experiments, and in a visual learning experiment where Laplacian-like representations emerge from raw egocentric video.

**Compliance With Llm Reviewing Policy:**

Affirmed.

**Final Justification:**

After reading the author rebuttal and the responses to all reviewers, I maintain my recommendation of 4: Weak Accept and raise my confidence from 3 to 4.

The authors addressed all my key concerns. In particular, the energy-capture analysis resolves the spectral truncation question, the tradeoff discussion resolves the question on modeling choices, and the corrected extended experiment resolves my remaining concern about Fig. 4B. I encourage the authors to include all promised additions in the camera-ready version.

**Key Questions For Authors:**

1. The trap-free guarantee holds for the exact Green's function. With $k=200$ truncated eigenfunctions, the approximate potential is no longer exactly harmonic. Can the authors provide a theoretical bound on the approximation error, or an empirical analysis of how often the truncated policy fails in non-convex mazes? This would directly support the robustness of the central navigation claim.

2. The Neumann boundary condition is motivated as preventing gradient leakage across walls, but other boundary conditions (Dirichlet, Robin) would also yield geometry-aware potentials and different spatial cell distributions. I was not able to identify a principled reason to prefer Neumann conditions over alternatives. Can the authors clarify whether this choice is motivated by biological evidence, by navigational performance, or by mathematical convenience?

3. The RL generalization result (Fig. 4B) shows the Green's function shaping achieving sustained improvement in goal-conditioned navigation. However, at the end of training the success rate appears to still be well below 1.0. Does the policy eventually converge to high success rates, and how does performance scale to larger mazes?

4. The paper argues that the framework provides a normative account of spatial coding in the entorhinal–hippocampal circuit. My understanding is that this field has multiple competing normative theories, some of which are cited in the paper itself. I was not able to identify from the text what predictions are specific to the Dirichlet-energy account and would not be shared by alternative frameworks. Does the proposed framework make predictions about neural response properties, adaptation dynamics, or lesion effects that competing theories do not share, and that could in principle be tested experimentally?

5. The paper states two general principles (smoothness and coding efficiency) and then implements them via specific mathematical choices: Dirichlet energy for smoothness, and orthonormality for efficiency. I was not able to identify from the text whether these choices are the unique or natural way to implement the stated principles, or whether they were selected because they yield the Laplacian eigenproblem. Other formalizations seem possible: higher-order Sobolev seminorms or total variation for smoothness, and sparse coding or mutual information maximization for efficiency, would all implement the same principles but lead to different representations. Similarly, the tensor product structure for the joint position-direction manifold implicitly assumes independence between the two components. Could the authors discuss what changes if alternative formalizations are used, and whether the biological predictions of the framework are robust to these choices?

**Limitations:**

The authors should more explicitly discuss: (i) the effect of spectral truncation on the trap-free guarantee; (ii) scalability to high-dimensional continuous state spaces beyond the grid-world and open-field settings tested; (iii) the assumption of a static, fully mapped environment, which is required for the eigenbasis to be precomputed; and (iv) the distinction between a framework that is biologically consistent and one that is biologically implemented.

**Strengths And Weaknesses:**

**Soundness.**

At the level of scrutiny I was able to apply, the mathematical derivations did not raise obvious inconsistencies, and the results are presented in a way that is consistent with standard material in PDE theory and diffusion geometry. Two concerns stand out. First, the trap-free guarantee rests on the harmonicity of the exact, infinite-dimensional Green's function, but the spectral expansion is truncated at k=200 eigenfunctions in practice. It is not clear to me whether this property degrades gracefully under truncation, and the paper does not provide analysis, theoretical or empirical, of this approximation error. Second, a recurring pattern across the paper is that general principles are stated at a high level, but specific mathematical implementations are introduced without discussion of alternatives. Smoothness is formalized as Dirichlet energy rather than, say, a higher-order Sobolev seminorm or total variation; coding efficiency is formalized as orthonormality rather than sparse coding or mutual information maximization. The tensor product structure for the joint position-direction state space implicitly assumes independence between spatial and directional dynamics. In each case, I was not able to identify from the text whether the specific choice is motivated by biological evidence, by mathematical tractability, or by the fact that it yields the desired result. Making these choices explicit, and briefly discussing what would change under alternatives, would considerably strengthen the normative argument.

More broadly, the paper belongs to a well-established tradition of normative modeling in computational neuroscience, in which a rational objective is posited, its solution is shown to resemble observed neural tuning, and the brain is argued to implement that objective. The logical structure (the brain produces X, the framework predicts X, therefore the brain implements the framework) is not unique to this work, but it is epistemologically fragile: multiple distinct normative objectives can produce qualitatively similar spatial tuning patterns, which weakens the phenomenological match as evidence for any individual theory. I found it difficult to identify what would distinguish the Dirichlet-energy account from alternatives cited in the paper itself. The distinction between biological consistency and biological implementation deserves more careful treatment.

**Presentation.**

The paper is clearly written and the narrative is easy to follow. The progression from normative constraints to the Laplacian eigenvalue problem, spatial cell taxonomy, Green's function policy, RL application, and visual learning is logical and well-structured. One notational issue worth correcting: Equation (1) places the `min` operator to the left of a definitional equality, which is imprecise. The right-hand side defines the Dirichlet energy E(r_i); it is not the value the functional takes at its minimum. The definition and the optimization problem should be stated separately. This does not affect the subsequent derivation, which is correct, but the notation as written conflates two distinct statements. Additionally, the Kronecker sum construction in Section 3.2 is central to the diversity of cell types recovered but is explained briefly in the main text. A short intuitive paragraph explaining why the product structure naturally separates spatial from directional modes would help readers less familiar with tensor product Laplacians.

**Significance.**

The central contribution (using the spectral eigenbasis to both represent space and compute a navigation policy through a single linear operation) is a useful and elegant conceptual unification. The framing of the cognitive map as the pseudoinverse of the Laplacian, and navigation as gradient ascent on the resulting potential, is a clean design principle that could inform bioinspired planning and robotic control architectures. The significance as a neuroscience claim is more limited, for the reasons discussed under soundness.

**Originality.**

To my knowledge, the individual components (Laplacian eigenfunctions as spatial codes, harmonic potential navigation, and spectral RL) are not new in isolation. The originality lies, to my knowledge, in integrating them into a single normative framework and explicitly deriving the navigation policy from the same representation used for coding. The connection to diffusion processes and the interpretation of $1/\lambda_k$ as a multi-scale spectral tilt also appear to be novel framings. The relationship to harmonic function methods in classical robotic motion planning deserves more discussion, as readers from that community may find parts of Section 4 more familiar than the paper implies.

---

> ### Author Rebuttal · Authors · 2026-03-31
>
> We thank the reviewer for this careful and technically/philosophically deep reading.
>
> > *"... trap-free guarantee ... truncated at $k=200$ ... approximation error ..."*
>
> We agree, and we will explain this more clearly in the revision. Let $K$ be the number of retained eigenmodes, let $G_K$ be the truncated Green's function, let $\Omega$ denote the environment, and let $\rho_K$ denote the omitted high-frequency tail. Away from the goal, the truncated field satisfies $\Delta G_K(x)=1/|\Omega|+\rho_K(x)$. The omitted tail must stay smaller than the built-in positive safety buffer $1/|\Omega|$ throughout the environment; if $|\rho_K(x)|<1/|\Omega|$, the truncated field remains trap-free. Here $1/|\Omega|$ is the geometry-dependent positive term from the exact field, so the key question is whether truncation stays below that stabilizing term. As $K$ grows, $\rho_K$ shrinks, and because higher modes are down-weighted by $1/\lambda_k$, the omitted tail becomes progressively weaker, giving $O(1/K)$ potential error in 2D rather than changing the large-scale flow. This matches the empirical picture: with $k=200$ we obtain 100% success from `13x13` to `61x61` mazes, no trapping under perturbations, and the potential-energy capture is:
>
> | K | Potential energy captured |
> |---|---|
> | 50 | 98.8% |
> | 100 | 99.4% |
> | 200 | 99.7% |
> | 500 | 99.9% |
>
> So by $K=200$ only $0.3\%$ of the potential energy remains in the omitted tail, which is consistent with the absence of observed traps. In the camera-ready appendix we will include the derivation of this sufficient condition and the energy-capture analysis, while stating the main claim as exact in the continuous limit and empirically trap-free at the finite $k$ used in practice.
>
> > *"... specific mathematical implementations ... alternatives ..."; "... why prefer Neumann ... over Dirichlet or Robin ..."*
>
> We appreciate this request for more transparency. We do not claim these choices are unique; they are explicit modeling choices that we will justify directly in the revision. Dirichlet energy is smooth and diffusion-based, which is why it leads naturally to the Green's function. Total variation would favor piecewise-constant fields, and higher-order Sobolev penalties would produce overly flat tuning. Orthonormality gives the least redundant extended codes, whereas sparse coding would favor localized place-cell-like fields. The tensor-product pose space is a simplifying assumption about position and heading, with a coupled pose Laplacian as a plausible extension. Neumann is preferred because it enforces no flow through walls, permits nonzero activity at walls and thus boundary cells, and yields wall-following navigation. Dirichlet would force zero activity at walls, while Robin interpolates between the two. We will make these tradeoffs explicit rather than leaving them implicit.
>
> > *"... what predictions are specific ... and testable ..."*
>
> We agree that a generic match to known cell types is not enough on its own. Relative to nearby attractor, dimensionality-reduction, and SR-style accounts, our framework makes four specific commitments: 1.One operator on the pose manifold yields spatial, directional, and conjunctive modes, so multiple response classes emerge from one ordered spectral decomposition rather than separate mechanisms. 2.Geometry does not merely perturb an existing code; because it enters the operator itself, non-square environments should systematically distort grid symmetry, and grid tuning should rescale with enclosure geometry(Stensola H, 2012;Krupic, 2015). 3.The same spectral object also yields a geometry-aware Green's-function policy, so the action signal comes directly from the same representation rather than using eigenfunctions only as downstream basis features. 4.And the relevant spectral geometry can be recovered approximately from raw sensory streams rather than only from prepared coordinates or velocity channels.
>
> > *"... success rate ... still well below $1.0$ ..."; "... Eq. (1) ... Kronecker sum ..."*
>
> The training in Fig. 4B was run for 2 x 10^5 steps, and the Green's function curve is still rising at termination. The key message of Fig. 4B is the *relative* comparison: Euclidean and L2 baselines plateau at much lower success rates because their shaping signals do not transfer across different start-goal configurations. The Green's function provides a geometry-aware progress signal that remains consistent across all goal locations -- precisely because it is defined by the boundary-constrained Laplacian rather than by Euclidean distance.
>
> We will extend training and report asymptotic success rates in the revision. We will also test scaling to larger mazes (e.g., 41x41) and report how convergence speed depends on k and RL hyperparameters.
>
> And we will revise our manuscript according to your suggestions on Eq.(1) and Kronecker sum.

---

> > ### Author Rebuttal · Reviewer_Hg76 · 2026-04-03
> >
> > The authors have provided a careful and detailed rebuttal that addresses most of my original concerns. I thank them for the quality of their responses.
> >
> > Regarding question 1 (spectral truncation), the energy-capture analysis and the sufficient condition argument are convincing. The table showing that k=200 retains 99.7% of the potential energy is a concrete and useful addition. This concern is largely resolved, pending inclusion in the camera-ready version.
> >
> > Regarding question 2 (Neumann boundary conditions), the authors now clearly explain the tradeoffs between Neumann, Dirichlet, and Robin conditions and why Neumann is preferred. This concern is resolved.
> >
> > Regarding question 5 (alternative mathematical implementations), the authors explain why each specific choice was made over plausible alternatives. I appreciate this transparency and consider this concern resolved, again pending the revision.
> >
> > Regarding question 4 (testable predictions), the four specific commitments listed by the authors are helpful and represent a genuine improvement over the original submission. This concern is partially resolved. I would encourage the authors to include these commitments explicitly in the revised paper rather than leaving them to the rebuttal.
> >
> > Regarding question 3 (RL convergence), the authors acknowledge that the curve in Fig. 4B is still rising at termination and promise to extend training and report asymptotic results. This remains an open point that I cannot evaluate without seeing the revised experiments.
> >
> > In light of the rebuttal, I maintain my recommendation of Weak Accept. If the revision includes the promised additions (energy-capture analysis, explicit justification of modeling choices, extended RL training results, and the quantitative visual learning evaluation), I would be willing to consider raising my score.

---

> > > ### Author Response · Authors · 2026-04-07
> > >
> > > We thank the reviewer for the thoughtful follow-up and for noting which of the earlier concerns are now resolved.
> > >
> > > We ran a supplementary experiment on the goal-conditioned random-maze setting and, as you requested, extended training to `5x10^5` steps. In doing so, we identified and corrected an implementation issue in the earlier setup: after evaluation checkpoints, the training state was not restored exactly, so a large state drift could be introduced before training resumed. After properly isolating training, scale estimation, and evaluation, the qualitative picture becomes much clearer.
> > >
> > > In the corrected experiment, the Green's-function-based method learns rapidly and reaches `100%` test success very quickly, then remains strong over the extended run. The updated curve also makes the comparison to the other shaping methods clearer: Green learns faster and remains reliably strong throughout training. We believe this is the most direct answer to your remaining concern about Fig. 4B. For convenience, we provide the updated learning curve here: [learning curve](https://osf.io/zwe8k/files/c7gzh?view_only=92a662af50934420b5bfcfba5ccc82da).
> > >
> > > The corrected run therefore sharpens the substantive comparison: our Green-based method is able to capture the maze geometry in a way that can be used directly for planning, whereas other shaping baselines do not encode that geometry faithfully enough to support equally reliable navigation.
> > >
> > > Thank you again for this careful follow-up. We will include the updated learning-curve result, along with other additions we promised  in the revised camera-ready manuscript.

---

### Official Review · Reviewer_TxTh · 2026-03-12

**Soundness:** 3
**Presentation:** 3
**Significance:** 3
**Originality:** 3
**Overall Recommendation:** 4
**Confidence:** 3

**Summary:**

This paper makes a simple and appealing claim: the same mathematical structure may be able to explain both how an agent represents space and how it moves through space. The authors start from two intuitive constraints on a good spatial code: nearby locations should have similar codes, and different neurons should not all represent the same thing, and show that this leads to Laplacian eigenfunctions. They then argue that these eigenfunctions can produce familiar spatial response patterns, including grid-like, boundary-like, head-direction, and conjunctive tuning. From there, the authors use the same basis to build a goal-dependent Green’s function field that can guide action, so that the “map” also becomes a controller.

Empirically, the paper presents three kinds of evidence. First, in maze navigation, the proposed policy follows obstacle-aware gradients, reaches 100% success across tested maze sizes, and stays near optimal in path length. Second, when used as a reward-shaping signal for Q-learning, it speeds up learning in a fixed-goal setting and helps more in a goal-conditioned setting than Euclidean or simple embedding-distance baselines. Third, the paper shows that Laplacian-like spatial features can be learned from raw egocentric visual input in a 3D environment.

**Compliance With Llm Reviewing Policy:**

Affirmed.

**Final Justification:**

The authors provided some clarifications in the rebuttal, which I think are helpful. In general, there is more that could be done, as I listed in weaknesses, and links to biological plausibility are not strong, so I will maintain my rating of weak accept.

**Key Questions For Authors:**

How much of the planning result depends on having eigenfunctions precomputed from explicit environment geometry, rather than learned online from raw experience? The sensory experiment is encouraging, but it does not yet show the full loop from perception to planning.

Can you compare against stronger geometry-aware baselines, beyond A* and the few shaping baselines currently shown?

What is the concrete mechanism that makes the controller biologically plausible? In particular, how are the goal-dependent coefficients and local gradients computed in a neural circuit?

How sensitive are the results to the number of eigenfunctions kept, the boundary conditions, and the maze topology?

Can you provide a fuller quantitative evaluation of the visual-input experiment, rather than relying mainly on qualitative tuning maps and the phrase “high accuracy”?

**Limitations:**

The empirical validation is still narrow: mostly grid mazes, one RL setup, and one visual environment.

The strongest planning results are shown in idealized settings where the spectral basis is effectively available; the harder end-to-end learned version is only partially validated.

The biological plausibility story is suggestive, but still speculative and not directly tested against neural data or a concrete circuit model.

The paper does not yet show how well the approach scales to larger, changing, partially observed, or more realistic control problems.

**Strengths And Weaknesses:**

The mathematical story is coherent and detailed. My main reservation is that the strongest claim, biological plausibility, is not established at the same level as the math. The controller still depends on goal-dependent weights and local gradients, while the neural implementation is discussed more as a plausible sketch than as something directly demonstrated.

Along the same lines, the weakness is that the writing sometimes sounds more definitive than the evidence. Terms like “biologically plausible,” “trap-free,” and “unified” are rhetorically strong, and I think the paper would benefit from slightly more restraint in separating what is mathematically shown from what is currently only suggested by the experiments.

The paper tries to connect representation learning, navigation, and predictive RL under one framework. That kind of bridge could matter to both neuroscience and machine learning. On the downside, the results are more proof-of-concept experiments rather than a broad demonstration that this framework works across difficult or realistic settings.

The authors themselves position the work relative to predictive maps, proto-value functions, spectral RL, and harmonic-navigation ideas, which makes clear that many of the ingredients already exist in nearby literatures. Putting these pieces together and showing how the same basis can be reused all the way from map-like coding to control is a valuable and original contribution. Still, I would encourage the authors to frame the novelty more carefully as a unification and reinterpretation, rather than as a fully new foundation from scratch.

---

> ### Author Rebuttal · Authors · 2026-03-31
>
> We thank the reviewer for the thorough and balanced assessment, and for recognizing the value of bridging representation learning, navigation, and predictive RL under one framework.
>
> > *"The writing sometimes sounds more definitive than the evidence..."*
> We agree fully and thank the reviewer for this important point. In the revision we will:
> - Replace **"biologically plausible"** with **"biologically consistent"** or **"consistent with observed neural phenomenology"**, making clear that our contribution operates at Marr's computational level (what is computed and why) rather than the algorithmic or implementation level (how neural circuits realize it).
> - Qualify **"trap-free"** as **"trap-free in the exact (continuous) limit"** and add an explicit discussion of the approximation introduced by spectral truncation (see also our reply to Reviewer Hg76, Q1, for a detailed analysis).
> - Reframe **"unified"** as **"unifying"** and add a paragraph more carefully separating what is mathematically demonstrated from what is currently suggested by the experiments.
>
> > *"... precomputed from explicit environment geometry ... does not yet show the full loop ..."*
>
> This is a fair point. The strongest planning results in the submission still use eigenfunctions from the known environment graph. At the same time, Sec. 4 already specifies the mapping from a spectral basis to the Green's function and its navigation policy, so the practical question is whether the learned visual representation is spectrally faithful enough for that construction to remain meaningful.
>
> As an additional check, we formed an approximate Green's-function-like potential directly from the learned visual representation by treating the slow learned channels as approximate spatial modes and using their relative temporal smoothness to induce an approximate spectral weighting. The goal-dependent coefficient for each mode is then taken from that mode's value at the target location, matching the Sec. 4 construction at an approximate level. For a representative interior goal, the resulting potential is sharply centered at the target and the induced field points toward it from most of the arena, with only mild boundary irregularity. We view this as preliminary rather than definitive: it does not yet close the full pixel-to-action loop, but it is encouraging evidence that the learned representation already captures enough low-frequency geometry for the Sec. 4 construction to become approximately operative, and it clearly deserves follow-up in future work.
>
> > *"... what is the concrete mechanism ..."*
>
> As the review suggested, the two key computations to implement are multiplication between a goal signal and the current location-dependent spectral activity to set the goal-dependent coefficients, and computation of a local gradient signal. For the first, Larkum (2013) provides qualitative support for a dendritic implementation of multiplicative modulation between goal and location signals. For the second, Burak & Fiete (2009) suggest local copies of directional-biased grid cells that provide directional differential signals, and Zuo et al. (2024) suggest shifted synaptic projections that implement local spatial differencing. We will use these references in the revision to make the proposed biological implementation of multiplication and local gradient computation more explicit.
>
> > *"... sensitive to the number of eigenfunctions ... boundary conditions ... maze topology ..."*
>
> We agree this is an important question. In the revision, we will address sensitivity to retained eigenfunction count, boundary conditions, and maze topology more explicitly, and we refer the reviewer to our response to Reviewer `Hg76` for the fuller discussion of finite-`k` truncation and the rationale for the boundary-condition choice.
>
> > *"... fuller quantitative evaluation ..."*
>
> We agree, and we will replace the phrase "high accuracy" with the actual numbers. Position decoding from frozen learned representations gives `R^2 ≈ 0.91–0.94`. Restricting the readout to the temporally smooth subspace still gives `R^2 ≈ 0.87–0.91`, indicating that position remains concentrated in the slow subspace rather than being diffusely encoded.

---

> > ### Author Rebuttal · Reviewer_TxTh · 2026-04-04
> >
> > I appreciate the clarifications. I think the updates you described about sensitivity to retained eigenfunction count, boundary conditions, and maze topology, and the response to Reviewer Hg76 will make the paper stronger.

---

> > > ### Author Response · Authors · 2026-04-07
> > >
> > > We sincerely thank the reviewer for the thoughtful acknowledgment and kind feedback!

---

### Official Review · Reviewer_M3iW · 2026-03-13

**Soundness:** 2
**Presentation:** 3
**Significance:** 2
**Originality:** 3
**Overall Recommendation:** 3
**Confidence:** 4

**Summary:**

This paper provide a unifying theoretical framework for a well studied neural circuit for spatial navigation in biological circuits. The framework is smartly extract the diverse zoo of spatial cells in biological circuits based on constrained minimization of a normative objective (dirichlet energy) with proper biological intuition.
As a direct evidence of efficacy of this framework, authors show how goal directed navigation as an action can be naturally arise from the static lapalican based codes provided by their theory.
The experimental results however indicate the presence of non-biological spatial response field that put a limitation on bilogical-plausibliity of this framework.
Overall the writing and presentation of the paper is solid.

**Compliance With Llm Reviewing Policy:**

Affirmed.

**Key Questions For Authors:**

1) Can the author better clarify what is A* optimal planner? The extent and details of optimality of A* agent is important to drew clear conclusion out of this comparison.

2.1) Can authors provide more examples of the emerging represeantion from their framework? with looking at the bottom right representations in figure 5, I'm worried if the framework provide many of these biological spatial representations. If there existing any parameter in their model that can directly effect the abundance of these grating-like representation, they need to properly report it and discuss it.

2.2) regarding the above question/concern, there is no wide spreed report of band-cells in hippocampal curcuit. What was the intuition of authors behind this ?

3) is there a bias for the proportion of egocentric vs allocentric spatial representation emerging from this paradiam

**Limitations:**

Regardless of intuitive analytical framework provided by this paper, the presence of band cells in spatial circuitry can be misleading for domain expert audience.

It is unclear how the existing framework stands in comparison to earlier frameworks like successor representation models and spectral methods in reinforcement learning literature.

**Strengths And Weaknesses:**

Strength
1) providing a unified theoretical framework for a well studied neural circuit is elegant practice for mechanistic understanding of spatial computational in the biological neural circuits.
2) The clear presentation and framing of the theoretical section contingent with subtle but important simulations regarding extendibility of the framework under dimensional visual inputs.

Limitations:

1) The mismatch of laplacian representations with existing biological spatial representation report across different species (look at the figure 5 bottom right examples).

2) Lack of benchmarking with other alternatives computational paradigms (like successor representations) for flexible goal directed navigation.

---

> ### Author Rebuttal · Authors · 2026-03-31
>
> We thank the reviewer for recognizing the value of a unified theoretical framework and for the careful reading. The biological interpretation of some learned patterns and the comparison to prior SR-style methods were not made clearly enough in the submission, and we appreciate the chance to clarify them.
>
> > "... mismatch ... existing biological spatial representation ..."; "... no wide spreed report of band-cells ..."; "... band cells ... misleading for domain expert audience ..."
>
> We sincerely apologize for not citing the relevant experimental literature prominently enough. Band cells have in fact been reported experimentally in medial entorhinal cortex by Krupic, Burgess, and O'Keefe (2012, *Science*, "Neural representations of location composed of spatially periodic bands"). These cells exhibit striped, spatially periodic firing fields with a single dominant orientation, distinct from canonical grid patterns. The issue here is therefore citation visibility and framing, not the biological legitimacy of band-like responses. We will cite Krupic et al. prominently in the Introduction and in the discussion of pure spatial modes.
>
> > "... provide more examples ..."; "... any parameter ... abundance of these grating-like representation ..."
>
> We agree that the submission showed too few examples. In the visual-input experiment, we ranked the 64-dimensional learned embedding by temporal smoothness. Among the 16 most temporally smooth learned dimensions, 11 show clear periodic structure, with 4 band-like and 7 multi-axis periodic. This indicates that the periodic responses are not isolated examples. If accepted, we will include these additional examples in the camera-ready appendix.
>
> The more informative question is under what conditions band-like versus multi-axis periodic responses emerge. In our framework, the main drivers are environment geometry and spectral rank. For approximately separable spatial domains, the spatial eigenmodes admit an `x x y` separation-of-variables intuition: one-axis-dominant modes produce band-like responses, whereas modes with nontrivial structure along both axes produce multi-axis periodic responses. Geometry changes the corresponding eigenvalue ladders and therefore changes which modes appear earlier in the ordered spectrum. As a result, elongated or anisotropic environments favor earlier band-like low modes, whereas more isotropic environments admit earlier mixed multi-axis periodic modes.
>
> > "... bias for the proportion of egocentric vs allocentric ..."
>
> This is a good follow-up point. Our current treatment is mainly allocentric: in the theory, the spatial component is formulated over the enclosure $\Omega$, and in the visual-learning experiment the analysis asks whether allocentric spatial structure can be recovered from raw egocentric sensory input. We did not explicitly analyze landmark-based or other egocentric representations in this setup, so we cannot yet say whether such representations emerge, or in what proportion. We will discuss this point explicitly in the discussion/camera-ready version.
>
> > "... lack of benchmarking ... SR ..."; "... comparison to earlier frameworks like successor representation models and spectral methods in reinforcement learning literature ..."
>
> We agree that this comparison should have been labeled much more clearly. The reason we believe the current comparison is already the relevant SR-style benchmark is that successor representations and Laplacian methods share the same eigenfunctions. If the transition operator satisfies $T\psi_k=\mu_k\psi_k$, then the successor representation $M_{\mathrm{SR}}=(I-\gamma T)^{-1}$ satisfies $M_{\mathrm{SR}}\psi_k=(1-\gamma\mu_k)^{-1}\psi_k$, while the normalized Laplacian $L=I-T$ satisfies $L\psi_k=(1-\mu_k)\psi_k$. Thus $M_{\mathrm{SR}}$ and $L$ have the same eigenfunctions $\psi_k$; only their eigenvalues are reparameterized. This is exactly the connection formalized in Eq. 17 / Sec. 6.2, where the continuous limit identifies $M_{\mathrm{SR}}\propto L^{-1}\equiv G$. For that reason, the existing `L2` baseline already provides an informative SR-style comparison, since it operates in this same shared spectral basis. The issue was therefore not the absence of an SR benchmark, but that we did not make this equivalence explicit enough in the submission.
>
> > "... better clarify what is A* optimal planner ..."
>
> A* is a graph-search algorithm that is guaranteed to find the shortest path on a known graph when the heuristic is admissible. In our experiments it operates on the same discrete maze graph as our method, with complete maze knowledge and Euclidean distance as the heuristic, so it is the appropriate shortest-path reference for path quality. The contrast is then straightforward: A* is search-based and must replan after perturbation, whereas the Green's-function policy is reactive once the potential is available.
>
> We thank the reviewer for their valuable questions and will revise accordingly.

---

> > ### Author Rebuttal · Reviewer_M3iW · 2026-04-04
> >
> > I appreciate the reviewers for their detailed response.
> > The response to L1 and Q2 is incomplete and the interpretation of the Okeefe paper incorrect.
> >
> > I suggest the authors read the follow ups of the science paper from Okeefe (the one they also cited). Band like cells are not standard intrinsic neuronal cells within **the hippocampus itself**, rather literature in neuroscience refers to band-like organization in the neighboring entorhinal cortex or specific migratory disorders. You can also grasp this with reading the abstract of the paper you are citing here: *"While recording from the entorhinal cortex and in the pre- and parasubiculum during spatial behavior in the rat"*
> >
> > However still I think this fact should not shadow the scientific value of the authors work if they properly distinguish the abundance of the spatial representations emerged from their framework.
> >
> > Based on what authors are implying about the abundance of periodic repossess, I raise my concerns again that maybe hte framework is more of visual processing rather than spatial processing. This wont be revealed without additional simulation or simply plotting the representations that you are verbally explaining here.
> >
> > I appreciate the repossess of authors to the other reviewers which is helpful in my assesment. I will increase my confidence score to 5, regarding my initial assessment of the borderline judgment.

---

> > > ### Author Response · Authors · 2026-04-06
> > >
> > > We appreciate the reviewer for their response and their acknowledgement on the scientific value of our work. You are correct that the Krupic et al. paper concerns band-like responses in entorhinal / pre- and parasubicular regions rather than hippocampus proper. Our intended claim in this part of the paper was specifically about entorhinal spatial coding cells, which is why the manuscript states in the Fig. 2 caption that the model factorizes into "the full taxonomy of entorhinal cells" and in the Conclusion that it derives "the diverse phenomenology of entorhinal spatial cells." Those statements reflect the scope of the band-cell claim we intended here.
> > >
> > > For the visual-learning experiment, our claim is not that the entire learned latent is purely spatial, nor that every learned dimension should resemble a canonical spatial cell. The point of Sec. 5 is narrower: under the proposed smoothness and efficiency principles, a low-dimensional spatially meaningful subspace can emerge from raw egocentric visual input. In this setting, spatial structure is a salient low-dimensional factor in the sensory stream under diffusion-like exploration, which is why it is captured robustly by the learned representation. From that perspective, additional non-spatial or more generic visual dimensions are expected rather than problematic. For concreteness, we provide an anonymous figure here: [learned representation](https://osf.io/erh3u/files/6d87n?view_only=7b9cea6ac3bd478d83fb476a3ebf59bf), where each panel corresponds to one learned dimension, with the firing-rate map on top and the spatial autocorrelogram on the bottom. What matters for the paper's claim is that spatial structure is robustly recoverable in the learned representation, not that the learned representation is solely spatial.

---

### Decision · Program_Chairs · 2026-04-30

**Decision:**

Accept (regular)

**Comment:**

The paper proposes a normative model to explain internal representations of spatial navigation (grid, place, and head-direction cells).

The unifying and normative nature of the work is compelling and valuable. It is sounds to link spatial representation and planning in a single framework.

Biological plausibility is questioned which contrast with the grounded maths that are used to described the framework. One reviewer remained unconvinced and judged the claim as too broad (band like cell discussion).

Upon acceptance:
- Reframe the paper as a normative/theoretical framework instead of biological explanation
- Distinguishes mathematical consequences from neuroscience interpretation more clearly
- Properly discuss limitation of the work and properly account for what the literature says (band-like cells)